# Fat Loss in Patients with Metastatic Clear Cell Renal Cell Carcinoma Treated with Immune Checkpoint Inhibitors

**DOI:** 10.3390/ijms24043994

**Published:** 2023-02-16

**Authors:** Ji Hyun Lee, Soohyun Hwang, ByulA Jee, Jae-Hun Kim, Jihwan Lee, Jae Hoon Chung, Wan Song, Hyun Hwan Sung, Hwang Gyun Jeon, Byong Chang Jeong, Seong Il Seo, Seong Soo Jeon, Hyun Moo Lee, Se Hoon Park, Ghee Young Kwon, Minyong Kang

**Affiliations:** 1Department of Radiology, Samsung Medical Center, Sungkyunkwan University School of Medicine, Seoul 06351, Republic of Korea; 2Department of Pathology and Translational Genomics, Samsung Medical Center, Sungkyunkwan University School of Medicine, Seoul 06351, Republic of Korea; 3Department of Urology, Samsung Medical Center, Sungkyunkwan University School of Medicine, 81 Irwon-ro, Gangnam-gu, Seoul 06351, Republic of Korea; 4Division of Hematology-Oncology, Department of Internal Medicine, Samsung Medical Center, Sungkyunkwan University School of Medicine, Seoul 06351, Republic of Korea; 5Department of Health Sciences and Technology, SAIHST, Sungkyunkwan University, Seoul 06351, Republic of Korea; 6Samsung Genome Institute, Samsung Medical Center, Seoul 06351, Republic of Korea

**Keywords:** immunotherapy, tumor biomarkers, kidney neoplasms, tumor microenvironment

## Abstract

The purpose of this study was to determine the prognostic impact of fat loss after immune checkpoint inhibitor (ICI) treatment in patients with metastatic clear cell renal cell carcinoma (ccRCC). Data from 60 patients treated with ICI therapy for metastatic ccRCC were retrospectively analyzed. Changes in cross-sectional areas of subcutaneous fat (SF) between the pre-treatment and post-treatment abdominal computed tomography (CT) images were expressed as percentages and were divided by the interval between the CT scans to calculate ΔSF (%/month). SF loss was defined as ΔSF < −5%/month. Survival analyses for overall survival (OS) and progression-free survival (PFS) were performed. Patients with SF loss had shorter OS (median, 9.5 months vs. not reached; *p* < 0.001) and PFS (median, 2.6 months vs. 33.5 months; *p* < 0.001) than patients without SF loss. ΔSF was independently associated with OS (adjusted hazard ratio (HR), 1.49; 95% confidence interval (CI), 1.07–2.07; *p* = 0.020) and PFS (adjusted HR, 1.57; 95% CI, 1.17–2.12; *p* = 0.003), with a 5%/month decrease in SF increasing the risk of death and progression by 49% and 57%, respectively. In conclusion, Loss of SF after treatment initiation is a significant and independent poor prognostic factor for OS and PFS in patients with metastatic ccRCC who receive ICI therapy.

## 1. Introduction

While the treatment paradigm for metastatic clear cell renal cell carcinoma (ccRCC) has undergone rapid evolution in the previous three decades, immune checkpoint inhibitors (ICIs) that target and block programmed death 1 (PD-1) or programmed death-ligand 1 (PD-L1) have resulted in further drastic shifts. Despite emerging treatment agents, a large percentage of patients do not respond, leading to early progression and poor prognosis [1]. Although PD-L1 expression, tumor mutational burden, and CD8+ tumor-infiltrating lymphocytes have been recognized as prognostic factors for ICI therapy, several studies on metastatic ccRCC have demonstrated conflicting results [2,3,4,5]. Thus, further studies are warranted to identify prognostic biomarkers and determine which patients are likely to respond.

Renal cell carcinoma (RCC) is a known malignancy in terms of the association between body mass index (BMI), adiposity, and prognosis, showing overall survival (OS) improvement in obese patients [6]. Despite the growing interest in this unexpected survival benefit of obesity, referred to as the “obesity paradox” [7,8], previous studies have focused on baseline features without considering their changes. Given that these changes, especially the rapid loss of adipose tissue, are also reported to have prognostic value in cancer patients [9,10,11], and that ICI therapy can lead to substantial changes in body composition [11], it may be reasonable to assume that these changes are also associated with patient prognoses. However, its prognostic impact in patients with metastatic ccRCC undergoing ICI therapy has not yet been established.

In this study, we aimed to determine the prognostic impact of fat loss determined using cross-sectional imaging after ICI treatment in patients with metastatic ccRCC. We also investigated the association between these changes and molecular features using targeted sequencing and RNA sequencing of tumor samples to understand the underlying biology.

## 2. Results

The study cohort consisted of 44 men and 16 women with a median age of 58 years (interquartile range (IQR), 52–65 years). The median intervals between the pre-treatment CT examination and treatment initiation, and that between the treatment initiation and post-treatment CT were 15.0 days (IQR, 7.0–28.5 days) and 61 days (IQR, 48.5–78.0 days), respectively. During the follow-up period, with a median of 14.1 months (IQR, 7.6–23.3 months), 26 patients (43.3%) died.

The median ΔSF, ΔVF, and ΔTF were −2.1%/month (IQR, −8.8–2.7%/month), 0.2%/month (−10.8–6.7%/month), and −0.4%/month (−6.7–3.5%/month), respectively. The optimal cut-off and corresponding log-rank *p* values for ΔSF, ΔVF, and ΔTF were −5%/month and <0.001, −8%/month and 0.056, and −10%/month and 0.003, respectively, defining SF loss, VF loss, and TF loss as ΔSF < −5%/month, ΔVF < −8%/month, and ΔTF < −10%/month, respectively. According to these criteria, the number of patients exhibiting SF loss, VF loss, and TF loss was 20 (33.3%), 46 (76.7%), and 13 (21.7%), respectively (Appendix A). From SF, VF, and TF, we adopted SF loss to stratify the patients, given that it had the lowest log-rank *p* value.

The baseline patient characteristics are shown in Table 1, which provides a comparison between patients with and without SF loss. Patients with SF loss were significantly younger (median, 53.5 years vs. 58.5 years, *p* = 0.031), had lower BMI (median, 21.9 kg/m^2^ vs. 24.0 kg/m^2^, *p* = 0.003) with higher prevalence of underweight (15.0% vs. 0.0%, *p* = 0.033) and lower prevalence of obesity (5.0% vs. 32.5%, *p* = 0.023), lower VFI (median, 23.1 cm^2^/m^2^ vs. 38.2 cm^2^/m^2^, *p* = 0.017) and TFI (median, 57.8 cm^2^/m^2^ vs. 87.7 cm^2^/m^2^, *p* = 0.042), compared with patients without SF loss. There were no other significant differences between the two groups, including the performance status and distribution of the IMDC risk criteria.

### 2.1. Overall Survival

In all patients, the median OS was 24.2 months (95% CI, 15.9–32.3 months). The OS was significantly shorter in patients with SF loss (median OS, 9.5 months; 95% CI, 7.2–15.9 months) and TF loss (median OS, 12.7 months; 95% CI, 3.6–15.9 months), compared with patients without SF loss (median OS, not reached; 95% CI, not estimated) (log-rank *p* < 0.001) or TF loss (median OS, 32.3 months; 95% CI, 17.3–32.3 months) (log-rank *p* = 0.003), respectively. Patients with VF loss (median OS, 12.7 months; 95% CI, 3.6–17.3 months) numerically showed a shorter OS compared to patients without VF loss that was not statistically significant (median OS, 32.3 months; 95% CI, 16.7–32.3 months) (log-rank *p* = 0.056) (Figure 1).

In univariable Cox proportional analysis, fat loss in terms of ΔSF, ΔVF, and ΔTF was significantly associated with poor OS. The association remained after adjustment for covariates, demonstrating a 49%, 15%, and 37% increased risk of death as SF, VF, and TF decreased by 5%/month. None of the baseline body composition features or ΔVSR were significantly associated with OS (Table 2).

### 2.2. Progression-Free Survival

In all patients, the median PFS was 9.4 months (95% CI, 3.3–23.8 months). PFS was significantly shorter in patients with SF loss (median PFS, 2.6 months; 95% CI, 1.7–4.4 months), VF loss (median PFS, 3.3 months; 95% CI, 2.1–9.4 months), and TF loss (median OS, 2.9 months; 95% CI, 1.8–4.4 months) compared with that in patients without SF loss (median PFS, 33.5 months; 95% CI, 9.5–37.7 months) (log-rank *p* < 0.001), VF loss (median PFS, 12.7 months; 95% CI, 3.7–33.9 months) (log-rank *p* = 0.034), or TF loss (median PFS, 12.9 months; 95% CI, 4.6–33.9 months) (log-rank *p* < 0.001) (Figure 2).

Univariable Cox proportional analysis showed that fat loss in terms of ΔSF and ΔTF was significantly associated with poor PFS. The association remained after adjustment for covariates, demonstrating a 57% and 36% increased risk of death as SF and TF decreased by 5%/month. However, the ΔVF, ΔVSR, and baseline body composition were not significantly associated with PFS (Table 3).

### 2.3. Response to Treatment

Treatment response was evaluable in 59 of the 60 patients. Complete response was achieved in six (10%) patients, partial response in 21 (35.0%) patients, and stable disease in nine (15.0%) patients, resulting in an ORR of 45.0% (95% CI, 32.1–58.4%). The number of patients achieving complete response, partial response, and stable disease in the first line ICI therapy group was four (15.4%), thirteen (50.0%), and three (11.5%), respectively, whereas in the subsequent nivolumab monotherapy group it was two (5.9%), eight (23.5%), and six (17.6%), respectively. Patients with SF loss had significantly lower ORR to ICI therapy compared to those without SF loss (10.0% vs. 62.5%, *p* < 0.001). This association was significant in both the first line (28.6% vs. 78.9%, *p* = 0.028) and subsequent therapy (0.0% vs. 47.6%, *p* = 0.005) subgroups. The best overall response differed between patients with and without SF loss, with patients with SF loss showing a lower partial response rate (10.0% vs. 47.5%, *p* = 0.004) and a higher progressive disease rate (75.0% vs. 20.0%, *p* < 0.001). Similarly, a significant difference in response observed between patients with and without SF loss, with patients with SF loss showing a lower rate of clinical benefit (10.0% vs. 62.5%, *p* < 0.001) and a higher rate of no clinical benefit (75.0% vs. 25.0%, *p* < 0.001) (Appendix A).

### 2.4. Interactions of Fat Loss with Line of Treatment and IMDC Risk Criteria

The association between fat loss in terms of ΔSF, ΔVF, ΔTF, and OS was not statistically significantly different between the first-line and subsequent therapy groups or between patients with favorable/intermediate and poor risk groups (*p* for interaction >0.05). Likewise, the association of fat loss with PFS did not significantly differ between the first-line and subsequent therapy groups, or between patients in the favorable/intermediate and poor risk groups (*p* for interaction >0.05) (Appendix A).

### 2.5. Characteristics of Genomic Alterations and Transcriptomes between Patients with and without SF Loss

In 42 patients for whom targeted sequencing data were available, we identified 17 recurrently altered genes. The most commonly altered genes in this cohort were *VHL* (*n* = 24, 57.1%), *PBRM1* (*n* = 14, 33.3%), *SETD2* (*n* = 11, 26.2%), and *BAP1* (*n* = 11, 26.2%), which are generally similar to those previously reported for ccRCC (Appendix A) [12,13]. There was no significant difference in recurrently mutated genes between samples with and without SF loss (*p* > 0.05). In addition, tumor mutational burden (TMB) and total indel count were similar between samples from patients with and without SF loss (*p* > 0.05) (Appendix A). Among the 60 patients in our study cohort, RNA sequencing data were available for 57. We identified 637 DEGs between patients with (*n* = 19) and without SF loss (*n* = 38) (Figure 3a, left). Upregulated genes in patients with SF loss were associated with cytokinesis and urogenital system development, whereas upregulated genes in patients without SF loss were associated with response to hypoxia, angiogenesis, and cytokine production (Figure 3a, right).

The proportion of immune cells calculated using CIBERSORTx is shown in Figure 3b. Notably, we found that the proportions of CD8 T cells and M1 macrophages were significantly lower in patients with SF loss than in those without SF loss (Figure 3c and Appendix A). We also observed that patients without SF loss had a higher proportion of active immune types, whereas patients with SF loss had higher rates of exhausted immune types (Figure 3d). Consistent with these findings, ssGSEA showed that angiogenesis (*p =* 0.003) and T-effector (*p =* 0.023) signatures from the IMmotion150 study [5] were downregulated in patients with SF loss compared with those without SF loss (Figure 4a,b). Moreover, immune-related gene signatures, including T cells (*p* = 0.039), NK cells (*p* = 0.045), and chemokines (*p* = 0.015), from the Javelin101 study [14] were downregulated in patients with SF loss compared to those without SF loss (Figure 4c,d). Additionally, the Th1 score, indicating anti-tumor-associated immunity, was also significantly decreased in patients with SF loss (*p <* 0.001) (Appendix A). IHC analysis of representative samples of patients with or without SF loss (*n* = 5 for each group) also demonstrated decreased number of CD8+ cells and granzyme B+ cells and lower PD-L1 CPS in patients with SF loss compared to those in patients without SF loss. However, no significant difference was observed in the CD68+ macrophage (pan-macrophage) count or PD-L1 TPS (Appendix A).

## 3. Discussion

Our analysis of patients with metastatic ccRCC treated with ICI therapy demonstrated that early fat loss after treatment is a significant prognostic factor affecting both OS and PFS. In addition, the associations between fat loss, OS, and PFS were independent of covariates and did not vary according to the line of treatment or the IMDC risk criteria. Although our results regarding the association between fat loss and poor prognosis are comparable to those of previous studies on patients undergoing ICI therapy [9,11], the strength of our study was that it showed that the transcriptomic features of the primary tumor samples differed according to fat loss, which may expand the understanding of the biological perspectives linking fat loss and poor prognosis. Transcriptomic features of primary tumor samples showed that patients with SF loss had an immune-suppressive tumor microenvironment (TME), including fewer CD8 T cells and M1 macrophage cells, a higher proportion of exhausted immune types, and downregulation of the immune-related gene signatures from the IMmotion150 [5] and Javelin101 studies [14] as well as anti-tumor immunity-related Th1 scores compared to those without SF loss.

Our study results also imply that the prognostic effect of subcutaneous and visceral fat differs and that loss of subcutaneous fat is more closely associated with poor prognosis. Since loss of subcutaneous fat could stratify patients according to OS, our analyses of transcriptomic features focused on its relationship with loss of subcutaneous fat. Although Han et al. [15] suggested different prognostic impacts of subcutaneous and visceral fat in patients with gastric cancer and cachexia by reporting that low subcutaneous fat was predictive of poor survival, whereas low visceral fat was not, the prognostic value of depletion of these two adipose tissues, their mechanisms, and whether they differ is not clear. In patients with non-small cell lung cancer, Degens et al. [9] argued that loss of subcutaneous and visceral fat is predictive of poor prognosis after nivolumab therapy. In a cohort of patients treated with ICI for various cancers, Crombe et al. [11] reported that the occurrence of subcutaneous adipopenia after treatment was correlated with a higher risk of progression. Similarly, Imai et al. [10] also reported that rapid depletion of subcutaneous fat indicates poor prognosis in hepatocellular carcinoma patients treated with sorafenib. Regarding the prognostic value of SF loss, our results are partially comparable to those of these previous studies. Given that subcutaneous fat is regarded as beneficial for lipid and glucose metabolism [16], depletion of energy reserves with exhaustion induced by cachexia may be a possible explanation for the worse prognosis in patients with SF loss. However, the association of SF loss with PFS and tumor response, as well as OS, indicates that factors other than nutritional or metabolic aspects, such as fat-tumor interaction, may be involved. Meanwhile, the time point at which post-treatment fat loss was evaluated in our study was similar to or earlier than in previous studies [10,11]. However, significant fat loss associated with poor prognosis has been reported to occur even at week 6 of treatment [9]. Therefore, further research is called for to find the ideal and earliest time to detect fat loss, and to elucidate how to improve the prognosis in patients showing early fat loss.

With growing interest in the obesity paradox, accumulating evidence suggests that the TME and its immunological perspective could play a critical role in this phenomenon. Obesity leads to T-cell modulation [17] and affects the balance between macrophage subtypes [18] that interact with tumor cells within the TME. In addition, obesity leads to T-cell dysfunction that is associated with enhanced antitumor efficacy [17] possibly through transcriptional and metabolic reprogramming [19], which has been considered convincing among several suggested hypotheses to address the obesity paradox in patients receiving ICI therapy. Obesity is believed to interact with immune checkpoint receptors in the TME, enhancing the response to ICI therapy and immune cell-mediated tumor regression through increased adipokine secretion from adipose tissue [17]. Notably, the proportional characteristics of immune cells observed in our patients without SF loss were similar to the results of previous studies that reported an increased CD8/CD4 ratio and favored the M1 over the M2 macrophage subtype in obesity [17,18]. Given that CD8 T cells have been recognized to contribute to antitumor immunity in patients with metastatic ccRCC undergoing nivolumab treatment [20] and that the high M2 macrophage subtype is associated with worse prognosis in patients with ccRCC [21], we believe that these different immune cell characteristics may explain why clinical outcomes differed between patients with and without SF loss in the present study. Future studies using imaging mass cytometry, which provides integrated spatial tissue analysis, may help to better understand tumor-immune interactions.

Our finding that patients without SF loss exhibit enrichment of angiogenesis signatures and upregulation of genes associated with response to hypoxia is also in line with a previous study by Sanchez et al. [8] which showed that obese patients harbor tumors with greater angiogenesis and peritumoral fat hypoxia. They also suggested that adipocyte hypertrophy leads to hypoxia and subsequent angiogenesis, simultaneously promoting tumor growth and tumor susceptibility to therapy. Although angiogenesis may explain the favorable outcomes in patients without SF loss in this study, there is still no conclusive evidence that angiogenesis is related to improved prognosis when ICI therapy is administered. Enhanced local delivery resulting from increased angiogenesis may be one possible explanation [8] and requires further investigation. Enrichment of chemokines in patients without SF loss may also explain the improved prognosis in these patients, as chemokines are known to participate in the anticancer immune response and inhibit cancer cell proliferation [22].

While the baseline body composition features were not significantly associated with prognosis in our study, patients who lost SF after treatment had a lower BMI and less adipose tissue at baseline. Therefore, the prognostic value of fat loss may be attributed to these baseline characteristics. However, considering that the baseline body composition features were not significantly associated with prognosis in our study, our results imply that their changes may have more prognostic value or at least partly affect prognosis. We also attempted to minimize the effect of reverse causality by adjusting for other relevant covariates. Nonetheless, since patients with SF loss had clinical features and TME status based on RNA sequencing data opposite to those of obese patients in previous studies [17,18], we presume that they may have already been associated with poor prognosis before they lost subcutaneous fat. Considering that obesity is related to TME modulation, the aforementioned association of baseline characteristics raises the suspicion that fat loss after treatment may also have further modulated the TME toward immune exhaustion and downregulation of immune-related signatures, leading to poorer prognosis. However, this causal relationship could not be determined in our study and remains a question as RNA sequencing data after treatment were not available. In this context, evaluating changes in TME status using post-treatment biopsies would also be an interesting topic worth investigating. Proteomic analysis of human blood-derived samples, which can be readily obtained before and after treatment, could be another attractive method to elucidate potential metabolic pathways.

Our study has some limitations. First, it was retrospectively conducted with a relatively small number of patients, which limits the power of the study. In addition, our study cohort was inhomogeneous and comprised both first-line ICI therapy and subsequent nivolumab monotherapy groups. Second, the causal relationship between fat loss and transcriptional characteristics could not be clearly determined. Third, residual confounding, despite multivariable analyses or unmeasured confounding factors, may also be possible. Fourth, the interval between pre-treatment and post-treatment cross-sectional imaging was inconsistent among patients. Fifth, in addition to the fact that RNA sequencing data were only available before treatment, our analysis of TME was based on bulk RNA sequencing, which may have obscured cell-specific transcriptomic features. Further studies using comprehensive single-cell profiling may provide a high-resolution view of cell-specific features, revealing novel TME characteristics [23].

Fat loss after ICI therapy was an independent prognostic factor for poor OS and PFS in patients with metastatic ccRCC. In particular, loss of subcutaneous fat, rather than visceral fat, is associated with poor prognosis. The transcriptomic characteristics, particularly immune-related features of the TME, significantly differed between patients with and without SF loss, implying that it may have the potential to link loss of fat and poor clinical outcomes in these patients. Further investigations are warranted to determine the causal relationship between fat loss and transcriptomic features to unravel how they affect patient survival.

## 4. Materials and Methods

### 4.1. Patients

This study was approved by our institutional review board (Samsung Medical Center, IRB file No. 2021-12-109). This study was conducted in accordance with the principles of the Declaration of Helsinki. We reviewed the electronic medical records of 84 consecutive patients with metastatic ccRCC, who received ICI therapy between November 2016 and April 2021. Among them, 76 patients with available pre-treatment abdominal computed tomography (CT) data were enrolled. After excluding patients with an interval between pre-treatment CT examination and treatment initiation exceeding 90 days (*n* = 4) and those who had not undergone post-treatment CT within 150 days after treatment initiation (*n* = 12), a total of 60 patients (33 treated with nivolumab monotherapy as a subsequent treatment after first-line tyrosine kinase inhibitor treatment, 27 treated with ICI-based therapy as a first-line treatment, 17 with nivolumab plus ipilimumab, 7 with pembrolizumab plus axitinib, and 3 with avelumab plus axitinib) were finally included in the analysis.

All patients received one of the following treatments: (i) nivolumab monotherapy (3 mg/kg intravenously, every 2 weeks, or 480 mg monthly); (ii) nivolumab (3 mg/kg) plus ipilimumab (1 mg/kg) every 3 weeks intravenously for four cycles, followed by nivolumab monotherapy (3 mg/kg every 2 weeks); (iii) pembrolizumab (200 mg intravenously, every 3 weeks) plus axitinib (5 mg orally, twice daily); or (iv) avelumab (10 mg/kg intravenously, every 2 weeks) plus axitinib (5 mg, orally, twice daily).

Patients continued to receive treatment until disease progression according to the Response Evaluation Criteria in Solid Tumors (RECIST) 1.1 criteria [24] was reached, or toxicity was unacceptable. Computed tomography (CT) or magnetic resonance imaging (MRI) examinations were performed every 3 or 4 months for tumor assessment.

### 4.2. Image Analysis

A board-certified radiologist (JHL) with seven years of experience in musculoskeletal imaging evaluated the body composition features, blinded to patient information. Pre- and post-treatment abdominal CT studies were analyzed using open-source semi-automated software (BMI_CT, version 1.0, Seoul, Republic of Korea; available at https://sourceforge.net/projects/muscle-fat-area-measurement/, accessed on 6 March 2022) based on MATLAB version R2010a (Mathworks Inc., Natick, MA, USA). At the level of the third lumbar vertebra [25], cross-sectional areas (cm^2^) of skeletal muscle (including the rectus, transverse and oblique abdominal muscles, psoas muscles, and paraspinal muscles), subcutaneous fat, and visceral fat were measured using a semiautomated method [26]. The total fat area was defined as the sum of the areas of subcutaneous and visceral fat. The visceral-to-subcutaneous fat ratio (VSR) was calculated by dividing the area of visceral fat by that of subcutaneous fat. The patients’ body composition areas (cm^2^) were normalized by dividing by the square of the height (m^2^) of the patient to calculate the skeletal muscle index (SMI) [27], subcutaneous fat index (SFI), visceral fat index (VFI), and total fat index (TFI) (cm^2^/m^2^). Changes in the cross-sectional areas of subcutaneous fat, visceral fat, total fat, and VSR between the pre-treatment and post-treatment CT images were expressed as percentages relative to the baseline measurements and were divided by the interval between the CT scans to calculate ΔSF, ΔVF, ΔTF, and ΔVSR (%/month), respectively.

### 4.3. Clinical Data Collection

Electronic medical records were reviewed to collect baseline demographic, clinical, and laboratory data at the time of starting the therapy of interest. Patients were characterized according to the International Metastatic Renal Cell Carcinoma Database Consortium (IMDC) risk criteria (anemia, thrombocytosis, neutrophilia, Karnofsky performance status <80%, time from diagnosis to treatment <1 year), categorizing them into favorable (score of 0), intermediate (score of 1–2), or poor (score of 3–6) groups [28]. BMI was calculated as the body weight (BW) divided by height squared (kg/m^2^) and categorized according to criteria for Asia–Pacific classification of underweight (<18.5 kg/m^2^), normal (18.5–22.9 kg/m^2^), overweight (23.0–24.9 kg/m^2^), or obese (≥25.0 kg/m^2^) [29]. ΔBW (%/month) was defined as change in BW between the pre-treatment and post-treatment CT divided by the interval between the CT scans.

The primary outcome was OS, which was calculated from the date of treatment initiation to death from any cause. Secondary outcomes included progression-free survival (PFS), objective response rate (ORR), and clinical benefits. PFS was defined as the time from the date of treatment initiation to the date of disease progression or death due to any cause. The ORR was defined as the proportion of patients who achieved partial or complete response. To assess clinical benefit, the response group was defined as follows: (i) clinical benefit: patients with complete response, partial response, or stable disease, if they had any objective reduction in tumor burden lasting at least 6 months; (ii) no clinical benefit: patients with progressive disease within 3 months; and (iii) intermediate benefit: all patients who do not fit into either clinical benefit or no clinical benefit [13]. Tumor response was assessed according to the RECIST Criteria 1.1 [24].

### 4.4. Targeted Sequencing and RNA Sequencing Preprocesses 

Targeted sequencing and RNA sequencing data from primary tumor specimens were collected with the patients’ written informed consent as part of another study approved by the Institutional Review Board (Samsung Medical Center, IRB file No. 2020-03-063).

To characterize the genomic landscape of metastatic ccRCC, we used targeted sequencing data to identify recurrently mutated genes in our cohort. Targeted sequencing of 380 cancer-related genes (CancerSCAN version 3.1, Seoul, Republic of Korea, a targeted-sequencing platform designed at Samsung Medical Center) was performed by extraction from formalin-fixed paraffin-embedded (FFPE) pre-treatment tumor tissues. Most samples had a mean coverage of ~900×, with coverage at hotspots well above the mean. Paired-end reads were aligned to the human reference genome (hg19) using BWA (v.0.7.5). SAMTOOLS (v0.1.18), GATK (v3.1-1), and Picard (v1.93) were used for file handling, local realignment, and the removal of duplicate reads, respectively. We recalibrated the base quality scores using GATK BaseRecalibrator based on known single-nucleotide polymorphisms (SNPs) and indels from dbSNP138.

Thereafter, we performed RNA sequencing to evaluate the transcriptome changes. Total RNA from FFPE pre-treatment tumor tissues was extracted using an RNA extraction kit (RNeasy Mini Kit, QIAGEN, Germantown, MD, USA), and RNA integrity was verified using a 2100 Bioanalyzer (Agilent, Santa Clara, CA, USA). Libraries for sequencing were generated using the QuantSeq 3′ Library Prep Kit (Lexogen Inc., Vienna, Austria) according to the manufacturer’s instructions and sequenced on a HiSeq 2000 system (Illumina, San Diego, CA, USA). The reads were mapped to the hg19 human reference genome using STAR with the default parameters. The number of reads mapped to each gene was calculated using RSEM. Data processing and analysis were performed using R/Bioconductor libraries. We analyzed gene ontology terms to characterize the biological roles of differentially expressed genes (DEGs) using the DAVID website.

### 4.5. Immune Cell Type, Immune Type, and ssGSEA (Single-Sample Gene Set Enrichment Test)

To evaluate immune-related characteristics, the proportion of immune cell types was calculated using CIBERSORTx [30]. Additionally, we evaluated immune types by classifying them into the active immune type and exhausted immune type, considering cancer progression and the patient’s clinical outcomes [31,32,33], using the Nearest Template Prediction (NTP) algorithm based on the expression of active stroma and normal stroma signatures [33]. To perform ssGSEA, we evaluated the association between published immune-related gene signatures and fat loss and identified overall expression patterns of immune-related gene signatures from IMmotion150 [5] and Javelin101 [14] studies. ssGSEA was computed using the “GSVA” package [34]. Additionally, the gene signatures of Th1 and Th2 were derived from Bindea et al.’s study [35].

### 4.6. Immunohistochemistry

Immunohistochemical (IHC) staining using whole section FFPE tissues was performed with the following primary antibodies: CD8 (clone 4B11, 1:400, Cat#NCL-L-CD8-4B11, Leica Biosystems, Newcastle, UK), Granzyme B (clone 11F1, 1:128, Cat#NCL-L-GRAN-B, Leica Biosystems, Newcastle, UK), CD68 (clone KP1, 1:2000, Cat#M0814, Dako, Glostrup, Denmark), and PD-L1 (clone 22C3, 1:50, Cat#M3653, Dako, Glostrup, Denmark). PD-L1 expression was evaluated as the tumor proportion score (TPS), which was defined as the percentage of tumor cells with PD-L1 expression, and the combined positive score (CPS), which was defined as the number of PD-L1-expressing tumors and immune cells (lymphocytes and macrophages) divided by the number of all tumor cells and multiplied by 100. To quantify CD8-, CD68-, and granzyme B-positive immune cells, up to 10 representative images of each IHC slide were acquired and analyzed by two expert genitourinary pathologists (S.H. and G.Y.K.) using the inForm 2.6 software (Akoya Biosciences, Marlborough, MA, USA). Positive cells were scored out of the total number of cells counted in representative images.

### 4.7. Statistical Analysis

Data are presented as absolute frequencies and percentages for categorical variables and medians and interquartile ranges for continuous variables. The optimal cut-off values to dichotomize ΔSF, ΔVF, and ΔTF were determined at the point that maximized the difference between OS in the two groups identified using the minimum log-rank *p* value approach [36]. Patients with ΔSF, ΔVF, and ΔTF below the cutoff values were regarded as having SF loss, VF loss, and TF loss, respectively. Patient characteristics were compared between subgroups stratified by fat loss; continuous variables were compared using the Mann–Whitney test, and categorical variables were compared using Fisher’s exact test.

The Kaplan–Meier method with the log-rank test was used to characterize the event-time distributions. A Cox proportional hazards regression model was used to explore whether changes in adiposity or baseline body composition features were associated with survival outcomes. Factors with *p* < 0.05 in the univariable tests were adjusted for age (≥65 years/<65 years), sex (male/female), ΔBW (continuously, per 1%/month), line of treatment (first line/non-first line), prior nephrectomy (yes/no), Eastern Cooperative Oncology Group (ECOG) performance status (≥1/0), IMDC risk criteria (favorable/intermediate/poor), and number of metastases (≥2/1) to determine whether their prognostic value was independent of these clinical covariates. The ORR, proportions of each best overall response, and response group according to changes in adiposity were compared using Fisher’s exact test. The interaction term in the Cox proportional hazard regression model was used to determine whether the association between the change in adiposity and OS or PFS differed across the line of treatment and IMDC risk criteria. Fisher’s exact test was used to evaluate whether gene-specific alterations, characteristics of genomic alterations, transcriptomes, and IHC results differed according to fat loss. DEGs were calculated by *t*-test between samples stratified by fat loss, and cut-off options were *p* < 0.05, and fold differences >1.0. Student’s *t*-test was performed for both groups.

All statistical analyses were performed using SPSS Statistics (version 27.0; SPSS Inc., Chicago, IL, USA), MedCalc^®^ Statistical Software (version 20.023; MedCalc Software Ltd., Ostend, Belgium), and R software (version 4.1.3; The R Foundation for Statistical Computing, Vienna, Austria). Statistical significance was set at *p* < 0.05.

## Figures and Tables

**Figure 1 ijms-24-03994-f001:**
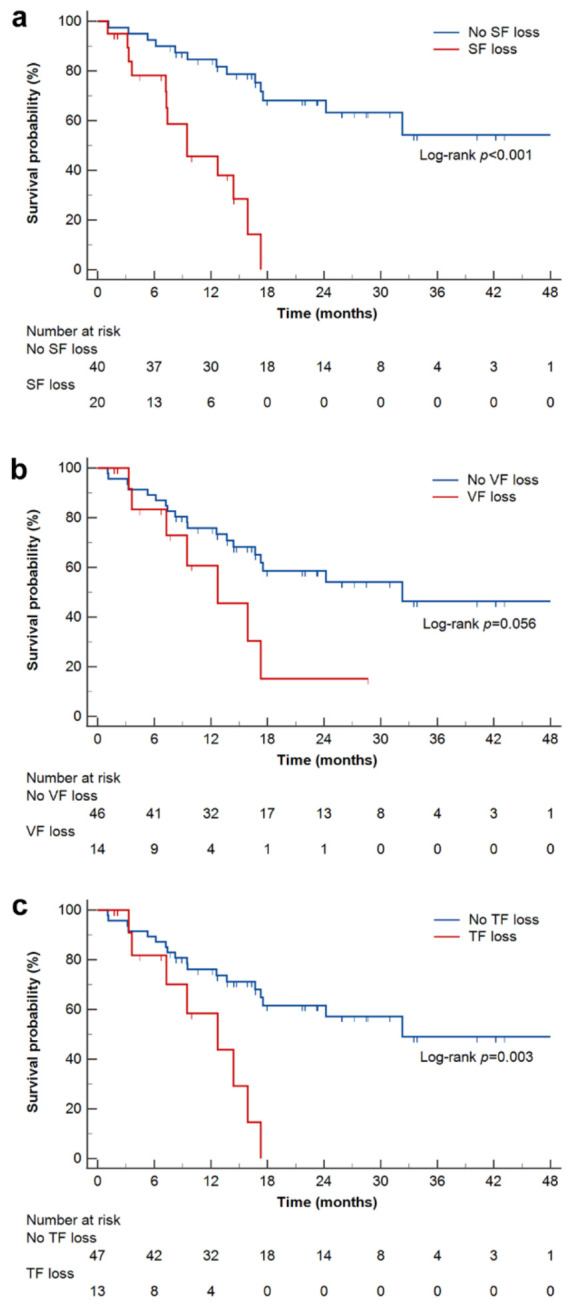
Kaplan–Meier estimates of overall survival, according to SF loss (**a**), VF loss (**b**), and TF loss (**c**). SF loss, VF loss, and TF loss were defined as ΔSF < −5%/month, ΔVF < −8%/month, and ΔTF < −10%/month, respectively. SF, subcutaneous fat; VF, visceral fat; TF, total fat.

**Figure 2 ijms-24-03994-f002:**
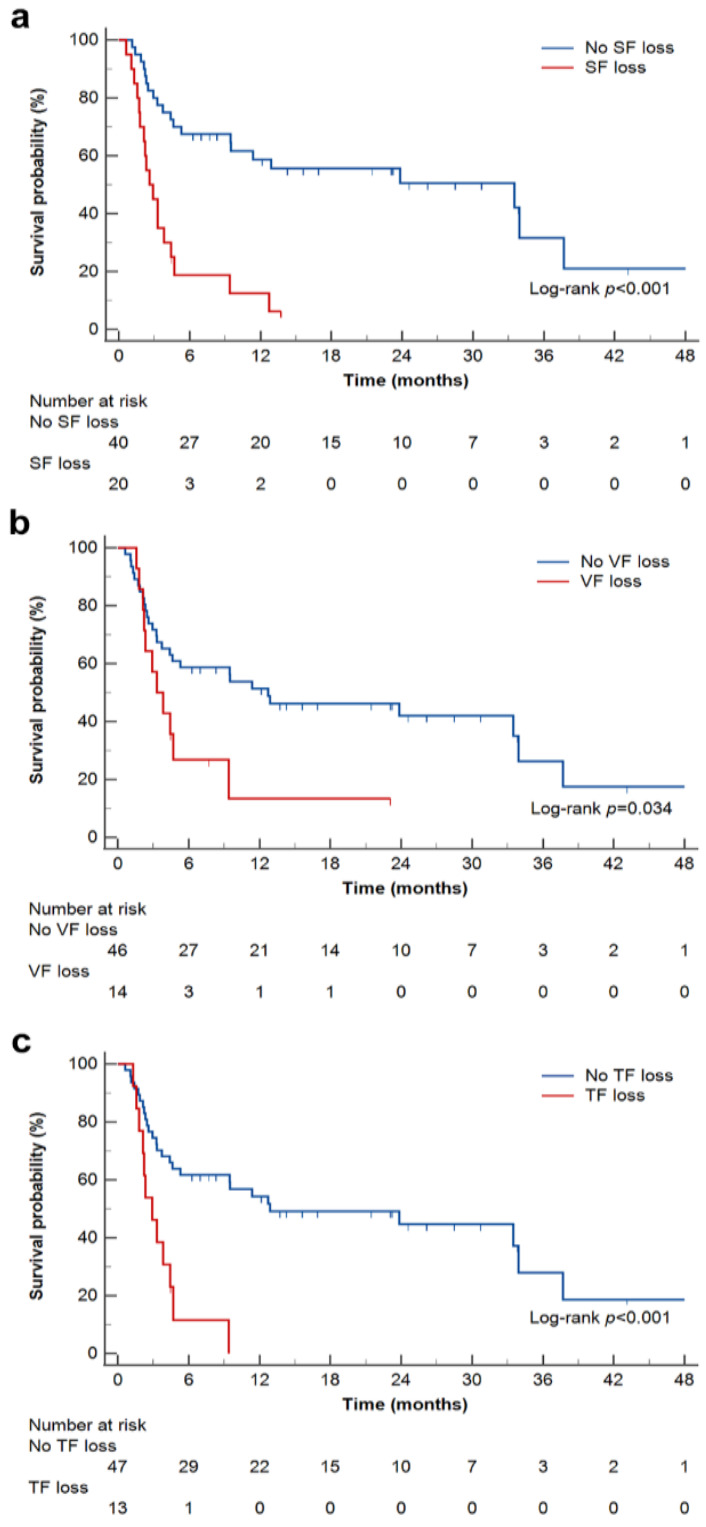
Kaplan–Meier estimates of progression-free survival, according to SF loss (**a**), VF loss (**b**), and TF loss (**c**). SF loss, VF loss, and TF loss were defined as ΔSF < −5%/month, ΔVF < −8%/month, and ΔTF < −10%/month, respectively. SF, subcutaneous fat; VF, visceral fat; TF, total fat.

**Figure 3 ijms-24-03994-f003:**
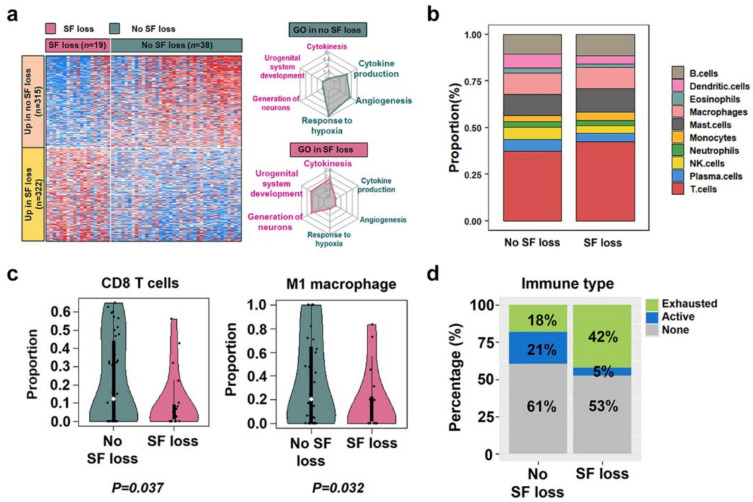
Transcriptomic characterization between samples from patients with and without SF loss. (**a**) A heatmap showing differentially expressed genes in each group stratified by SF loss (**left**). Radar plots showing enriched gene ontology terms using upregulated genes in samples from patients without SF loss (**top right**) and with SF loss (**bottom right**). The increasing color gradient from blue to white to red corresponds with the increasing expression values. (**b**) CIBERSORTx showing the proportion of distinct immune cell subpopulations according to SF loss. (**c**) Comparison of the proportion of CD8 T cells (**left**) and the proportion of M1 macrophages (**right**). (**d**) Barplots showing the percentage of immune types, including active-immune, exhausted-immune, and non-immune types in groups with and without SF loss. Green-colored, navy-colored, and gray-colored areas represent the exhausted-immune type, active-immune type, and non-immune type, respectively. SF, subcutaneous fat; GO, gene ontology.

**Figure 4 ijms-24-03994-f004:**
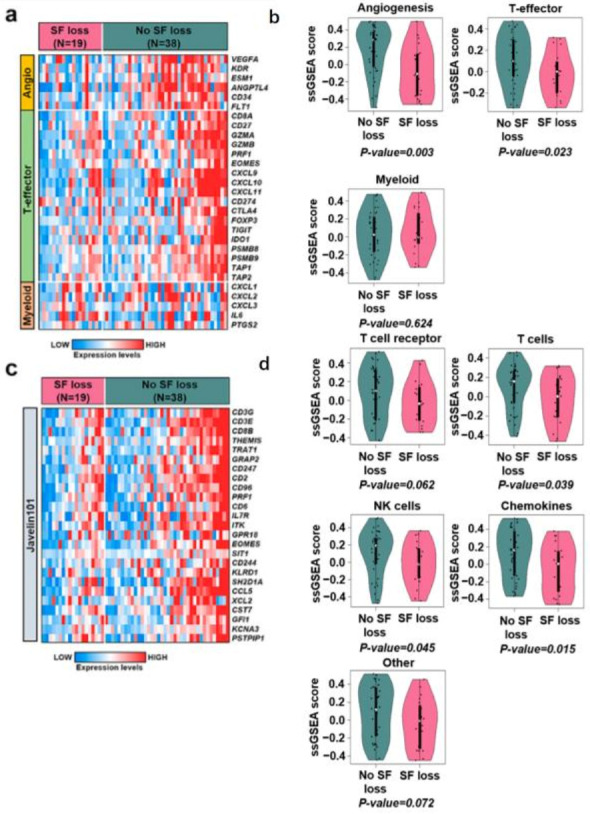
Immune-related gene signatures between samples from patients with and without SF loss. (**a**) A heatmap showing immune-related genes from the IMmotion150 study. The increasing color gradient from blue to white to red corresponds with the increasing expression value. (**b**) Violin plots showing ssGSEA scores using gene signatures from the IMmotion150 study. (**c**) A heatmap showing immune-related genes from the Javelin101 study. (**d**) Violin plots showing ssGSEA score using gene signatures from the Javelin101 study. SF, subcutaneous fat; ssGSEA, single-sample gene set enrichment test.

**Table 1 ijms-24-03994-t001:** Baseline characteristics.

Variables	Total (*n* = 60)	SF Loss ^‡^ (*n* = 20)	No SF Loss (*n* = 40)	*p*
Age (years) *	58.0 [52.0, 65.0]	53.5 [43.5, 62.0]	58.5 [54.0, 66.5]	0.031
Male sex ^†^	44 (73.3%)	13 (65.0%)	31 (77.5%)	0.360
BMI (kg/m^2^) *	22.8 [21.0, 24.9]	21.9 [19.4, 23.0]	24.0 [21.9, 25.6]	0.003
Underweight (<18.5) ^†^	3 (5.0%)	3 (15.0%)	0 (0.0%)	0.033
Normal (18.5–22.9) ^†^	28 (46.7%)	12 (60.0%)	16 (40.0%)	0.176
Overweight (23.0–24.9) ^†^	15 (25.0%)	4 (20.0%)	11 (27.5%)	0.753
Obese (≥25.0) ^†^	14 (23.3%)	1 (5.0%)	13 (32.5%)	0.023
SMI (cm^2^/m^2^) *	46.9 [38.7, 51.8]	44.5 [37.1, 48.1]	47.3 [39.4, 52.3]	0.227
SFI (cm^2^/m^2^) *	36.5 [29.9, 52.5]	33.8 [28.4, 45.2]	38.1 [13.4, 56.8]	0.218
VFI (cm^2^/m^2^) *	29.7 [14.4, 54.1]	23.1 [6.3, 32.8]	38.2 [18.0, 62.5]	0.017
TFI (cm^2^/m^2^) *	74.3 [46.8, 98.6]	57.8 [40.5, 81.6]	87.7 [54.1, 108.9]	0.042
VSR *	0.71 [0.44, 1.11]	0.63 [0.18, 0.97]	0.77 [0.55, 1.31]	0.083
Treatment agent ^†^				0.647
Nivolumab monotherapy ^†^	33 (55.0%)	13 (65.0%)	20 (50.0%)	0.409
Nivolumab + ipilimumab ^†^	17 (28.3%)	5 (25.0%)	12 (30.0%)	0.769
Pembrolizumab + axitinib ^†^	7 (11.7%)	2 (10.0%)	5 (12.5%)	1.000
Avelumab + axitinib ^†^	3 (5.0%)	0 (0.0%)	3 (7.5%)	0.544
ECOG PS ≥ 1, *n* ^†^	36 (60.0%)	15 (75.0%)	21 (52.5%)	0.161
Non-first line treatment ^†^	34 (56.7%)	13 (65.0%)	21 (52.5%)	0.416
Prior nephrectomy ^†^	44 (73.3%)	13 (65.0%)	31 (77.5%)	0.360
IMDC risk criteria ^†^				0.057
Favorable ^†^	6 (10.0%)	0 (0.0%)	6 (15.0%)	0.165
Intermediate ^†^	34 (56.7%)	10 (50.0%)	24 (60.0%)	0.582
Poor ^†^	20 (33.3%)	10 (50.0%)	10 (25.0%)	0.081
NLR *	3.38 [1.90, 5.08]	4.11 [2.90, 6.60]	2.96 [1.86, 4.47]	0.119
Number of metastases ≥2 ^†^	44 (73.3%)	16 (80.0%)	28 (70.0%)	0.541

BMI: body mass index; SMI: skeletal muscle index; SFI: subcutaneous fat index; VFI: visceral fat index; TFI: total fat index; VSR, visceral-to-subcutaneous fat ratio; ECOG PS: Eastern Cooperative Oncology Group performance status; IMDC: International Metastatic Renal Cell Carcinoma Database Consortium; NLR, blood neutrophil-to-lymphocyte ratio; SF: subcutaneous fat. * Mann-Whitney test. Data are presented as medians and interquartile ranges in square brackets. ^†^ Fisher’s exact test. Data are the number of patients and percentiles in parentheses. ^‡^ Defined as ΔSF < −5%/month.

**Table 2 ijms-24-03994-t002:** Results of Cox regression analysis for overall survival.

Variables	Univariable		Multivariable ^§^	
	HR (95% CI)	*p*	Adjusted HR (95% CI)	*p*
BMI *	0.82 (0.55–1.25)	0.361	-	-
SMI ^†^	0.75 (0.46–1.20)	0.224	-	-
SFI ^†^	1.12 (0.92–1.35)	0.258	-	-
VFI ^†^	0.98 (0.84–1.14)	0.783	-	-
TFI ^†^	1.02 (0.92–1.12)	0.727	-	-
VSR ^‡^	0.98 (0.91–1.05)	0.523	-	-
ΔSF ^⁋^	1.60 (1.20–2.13)	0.001	1.49 (1.07–2.07)	0.020
ΔVF ^⁋^	1.19 (1.02–1.39)	0.026	1.15 (1.00–1.32)	0.043
ΔTF ^⁋^	1.48 (1.16–1.89)	0.002	1.37 (1.05–1.77)	0.019
ΔVSR ^⁋^	1.06 (0.92–1.22)	0.416	-	-

BMI: body mass index; SMI: skeletal muscle index; SFI: subcutaneous fat index; VFI: visceral fat index; TFI: total fat index; VSR, visceral-to-subcutaneous fat ratio; SF: subcutaneous fat; VF: visceral fat; TF: total fat; HR: hazard ratio; CI: confidence interval. * Continuously, per 3 kg/m^2^. ^†^ Continuously, per 10 cm^2^/m^2^. ^‡^ Continuously, per 0.1. ^⁋^ Continuously, per −5%/month. ^§^ Adjusted for age (≥65 years/<65 years), sex (male/female), ΔBW (continuously, per 1%/month), line of treatment (first line/non-first line), prior nephrectomy (yes/no), ECOG performance status (≥1/0), IMDC risk criteria (favorable/intermediate/poor), and number of metastases (≥2/1).

**Table 3 ijms-24-03994-t003:** Results of Cox regression analysis for progression-free survival.

Variables	Univariable		Multivariable ^§^	
	HR (95% CI)	*p*	Adjusted HR (95% CI)	*p*
BMI *	1.13 (0.79–1.60)	0.508	-	-
SMI ^†^	1.25 (0.86–1.82)	0.248	-	-
SFI ^†^	1.14 (0.97–1.34)	0.116	-	-
VFI ^†^	1.00 (0.89–1.13)	0.998	-	-
TFI ^†^	1.03 (0.95–1.13)	0.459	-	-
VSR ^‡^	0.98 (0.92–1.03)	0.427	-	-
ΔSF ^⁋^	1.50 (1.16–1.95)	0.002	1.57 (1.17–2.12)	0.003
ΔVF ^⁋^	1.09 (0.98–1.21)	0.101	-	-
ΔTF ^⁋^	1.30 (1.07–1.58)	0.009	1.36 (1.11–1.66)	0.003
ΔVSR ^⁋^	1.04 (0.93–1.15)	0.510	-	-

BMI: body mass index; SMI: skeletal muscle index; SFI: subcutaneous fat index; VFI: visceral fat index; TFI: total fat index; SF: subcutaneous fat; VF: visceral fat; TF: total fat; HR: hazard ratio; CI: confidence interval. * Continuously, per 3 kg/m^2^. ^†^ Continuously, per 10 cm^2^/m^2^. ^‡^ Continuously, per 0.1. ^⁋^ Continuously, per −5%/month. ^§^ Adjusted for age (≥65 years/<65 years), sex (male/female), ΔBW (continuously, per 1%/month), line of treatment (first line/non-first line), prior nephrectomy (yes/no), ECOG performance status (≥1/0), IMDC risk criteria (favorable/intermediate/poor), and number of metastases (≥2/1).

## Data Availability

The datasets used and/or analyzed during this study are available from the corresponding author upon reasonable request.

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
