# Peer review of "Fat Loss in Patients with Metastatic Clear Cell Renal Cell Carcinoma Treated with Immune Checkpoint Inhibitors"

_ijms, 2023, doi:10.3390/ijms24043994_

Round 1

Reviewer 1 Report

It is known that fats are associated with cancer immunity.

In this study, the authors use immune checkpoint inhibitor in the treatment of mRCC and examined whether Fat loss in the visceral fat or subcutaneous fat was related to the prognosis. They found fat loss of subcutaneous fat is related to poor prognosis. Therefore, the authors performed RNA analysis between groups with and without reduced subcutaneous fat and found differences between the two groups.

Limitation of this study

1.       Retrospective

2.       Small number of patients

3.       The RNA sequencing data were available only before treatment.

I have several questions.     

1. TME of SF loss group is immune-suppressive before treatment. It may be the cause of poor prognosis.  It is necessary biomarker to know TME before treatment or early after initiation of treatment. In this study, the post treatment CT was performed 61 days (48.5-78) after treatment initiation. In the checkmate 214 study, The PFS curve was worsen 2-6 months after initial treatment. To use fat loss as a biomarker, I think the post treatment CT was performed earlier. How do you think? Is it possible to detect the fat change using CT taken earlier than this study(for example, 2ws or 1month later treatment initiation). Please discuss about it.

2.       There were several reports that SF/VF ratio was associated with the prognosis of esophageal or endometrial cancer. Do you examine SF/VF ratio before treatment and the change of SF/VF ratio?

Author Response

Q1. TME of SF loss group is immune-suppressive before treatment. It may be the cause of poor prognosis.  It is necessary biomarker to know TME before treatment or early after initiation of treatment. In this study, the post treatment CT was performed 61 days (48.5-78) after treatment initiation. In the checkmate 214 study, The PFS curve was worsen 2-6 months after initial treatment. To use fat loss as a biomarker, I think the post treatment CT was performed earlier. How do you think? Is it possible to detect the fat change using CT taken earlier than this study(for example, 2ws or 1month later treatment initiation). Please discuss about it.

REPLY) We fully agree that early detection of poor prognostic factor (fat loss in our study) should be desirable. Considering that previous studies evaluated body composition change after treatment evaluated it after 2-4 months after treatment, we believe that the timepoint in our study (2 months) could be considered ‘relatively early’. However, as there have been another study that described significant fat loss occurred even at week 6 of treatment, it may be reasonable to presume that more early detection of fat loss is possible. We additionally described this important issue in the discussion section.

  1. There were several reports that SF/VF ratio was associated with the prognosis of esophageal or endometrial cancer. Do you examine SF/VF ratio before treatment and the change of SF/VF ratio?

REPLY) Thank you for suggesting this useful metric in body composition assessment. . Please refer to Table 2 and 3, where VSR and ΔVSR have been analyzed additionally.

Reviewer 2 Report

This study represent a significant amount of work that must me recognized

First you uses clinical and survival data to explore relationships between fatloss and clinical outcomes

You then used an interseting dataset  with transcriptomics, genomic alterations and cibersort to explore relationship between fatloss and cancer cell features and tumor microenvironment.

However some issues need to be discussed : 

1. To demonstrate the interest of fatloos as a relevant clinical prognostic factor : 

1.1/ IMDC should have been in the multivariate analyses for it is the main confounding prognostic factor, (you have an enriched poor IMDC prognostic population among patients with fatloss).

1.2/ Also weightloss (if you have the data) should be assessed because it can be a simpler surrogate for fatloss as a prognosis factor

1.3/ In the discussion, you should question the applicability in clinical pratice and the main obstacles that need to be overcome so fatloss can be a relevant information (for example, ideal or minimum timing to evaluate fatloss during treatment is not known and should be validated in a prospective cohort, or what could be done with this information :  is nutritional support expected  to be helpful or not ? could we consider to intensify treatment in case of fatloss ? etc)

1.4 :  minor remark : do you have data for C reactive protein levels or absolute neutrophil count at baseline ? i suspect cancers with fatloss to be associated with inflammation pathways.

2.  About the molecular analyses : 

I assume the molecular analyses used  and also the gene panels were not dedicated only to this work. I still remains a rather and impressive work but discussion should open on what should be done to understand better the link between fatloss and cancer molecular features or TME (inflammation or metabolic pathway to be explored ?)

Author Response

Q1. IMDC should have been in the multivariate analyses for it is the main confounding prognostic factor, (you have an enriched poor IMDC prognostic population among patients with fatloss).

REPLY) Please refer to “Statistical Analysis” of M&M section and the footnotes of Tables 2 and 3 where we clarified that IMDC risk criteria has been included in the multivariable analyses.

Q2. Also weightloss (if you have the data) should be assessed because it can be a simpler surrogate for fatloss as a prognosis factor

REPLY) We totally agree to your brilliant opinion that body weight is an essential biomarker that can be easily assessed. However, the prognostic impact of weight loss in cancer patients has already been widely investigated in patients with cancer including renal cell carcinoma (PMID: 33951326, 28655483, 29693333). Although it seems reasonable that weight loss in our study cohort also could be associated with poor prognosis, the focus of our study was to explore the impact of body composition features and its changes that could not be assessed by BMI or body weight. Most importantly, we directly measured body composition features with cross-section imaging, as BMI cannot differentiate between the body composition features (water, fat, muscle, ect.). In particular, we emphasized the relationship between fat metabolism, ICI therapy, and tumor microenvironment, as described in the introduction and discussion section.

Q3. In the discussion, you should question the applicability in clinical pratice and the main obstacles that need to be overcome so fatloss can be a relevant information (for example, ideal or minimum timing to evaluate fatloss during treatment is not known and should be validated in a prospective cohort, or what could be done with this information :  is nutritional support expected  to be helpful or not ? could we consider to intensify treatment in case of fatloss ? etc)

REPLY) We also acknowledge that there are several obstacles that should be overcome to use fat loss as a useful biomarker for cancer patients in daily clinical practice. In particular, we totally agree that the ideal and earliest timepoint to assess body composition change after treatment should be validated. How to prevent worse outcomes in patients who shows fat loss could be another important and interesting issue. We described this matter in the discussion section, hoping that future study could be performed.

Q4. minor remark : do you have data for C reactive protein levels or absolute neutrophil count at baseline ? i suspect cancers with fatloss to be associated with inflammation pathways.

REPLY) (median values; 4.11 in SF loss group, 2.96 in no SF loss group, p=0.119) please refer to Table 1. We also hope your understanding that we did not adjust for NLR additionally, because of its statistical insignificance as well as small number of patients. In addition to the previously reported relationship between serum inflammatory markers and tumor microenvironment (PMID: 27665104, 24982357), we believe that our study additionally revealed close association between fat metabolism and tumor microenvironment, from a new perspective different from the previous studies.

Q5.  About the molecular analyses : 

REPLY) Thank you for pointing out this important issue. Although we investigated TME to link fat loss to poor clinical outcomes, we acknowledge that more dedicated analyses may help understanding to link fat loss to tumor progression and treatment response further. In addition to single-cell profiling that already have been described in the limitation section, we additionally discussed potential ways to further elucidate their relationships.

Round 2

Reviewer 1 Report

Thank you for your revised version.

I think it is acceptable.